# Perturbation of Methionine/S-adenosylmethionine Metabolism as a Novel Vulnerability in MLL Rearranged Leukemia

**DOI:** 10.3390/cells8111322

**Published:** 2019-10-25

**Authors:** Aditya Barve, Alexis Vega, Parag P. Shah, Smita Ghare, Lavona Casson, Mark Wunderlich, Leah J. Siskind, Levi J. Beverly

**Affiliations:** 1Department of Pharmacology and Toxicology, University of Louisville, Louisville, KY 40202, USA; aditya.barve@louisville.edu (A.B.); Leah.Siskind@louisville.edu (L.J.S.); 2Department of Biochemistry and Molecular Genetics, University of Louisville, Louisville, KY 40202, USA; Alexis.Vega@louisville.edu; 3James Graham Brown Cancer Center, University of Louisville, Louisville, KY 40202, USA; parag.shah@louisville.edu; 4Department of Medicine, University of Louisville, Louisville, KY 40202, USA; Smita.Ghare@louisville.edu (S.G.); Lavona.Casson@Louisville.edu (L.C.); 5Department of Experimental Hematology and Cancer Biology, Cincinnati Children’s Hospital Medical Center, Cincinnati, OH 45229, USA; Mark.Wunderlich@cchmc.org

**Keywords:** AML, Methionine, SAM, mouse model, MLL

## Abstract

Leukemias bearing mixed lineage leukemia (MLL) rearrangement (MLL-R) resulting in expression of oncogenic MLL fusion proteins (MLL-FPs) represent an especially aggressive disease subtype with the worst overall prognoses and chemotherapeutic response. MLL-R leukemias are uniquely dependent on the epigenetic function of the H3K79 methyltransferase DOT1L, which is misdirected by MLL-FPs activating gene expression, driving transformation and leukemogenesis. Given the functional necessity of these leukemias to maintain adequate methylation potential allowing aberrant activating histone methylation to proceed, driving leukemic gene expression, we investigated perturbation of methionine (Met)/S-adenosylmethionine (SAM) metabolism as a novel therapeutic paradigm for MLL-R leukemia. Disruption of Met/SAM metabolism, by either methionine deprivation or pharmacologic inhibition of downstream metabolism, reduced overall cellular methylation potential, reduced relative cell numbers, and induced apoptosis selectively in established MLL-AF4 cell lines or MLL-AF6-expressing patient blasts but not in BCR-ABL-driven K562 cells. Global histone methylation dynamics were altered, with a profound loss of requisite H3K79 methylation, indicating inhibition of DOT1L function. Relative occupancy of the repressive H3K27me3 modification was increased at the DOT1L promoter in MLL-R cells, and DOT1L mRNA and protein expression was reduced. Finally, pharmacologic inhibition of Met/SAM metabolism significantly prolonged survival in an advanced, clinically relevant patient–derived MLL-R leukemia xenograft model, in combination with cytotoxic induction chemotherapy. Our findings provide support for further investigation into the development of highly specific allosteric inhibitors of enzymatic mediators of Met/SAM metabolism or dietary manipulation of methionine levels. Such inhibitors may lead to enhanced treatment outcomes for MLL-R leukemia, along with cytotoxic chemotherapy or DOT1L inhibitors.

## 1. Introduction

Mixed lineage leukemia 1 (MLL) gene rearrangement is a defining feature of a unique group of particularly aggressive and chemotherapy resistant acute leukemias. Rearrangement of the MLL gene results in fusion to a variety of partner genes, with approximately 80% occurring with one of six common gene partners (see below). MLL-rearrangement (MLL-R) is detectable in 10% of leukemia cases and have the overall worst prognosis among cytogenetically abnormal leukemias [1]. MLL-R causes the aberrant expression of oncogenic MLL fusion proteins (MLL-FPs), and disease typically manifests as either acute myeloid or acute lymphoid leukemias (AML or ALL) and accounts for 10% of adult AML cases and 70–80% of infant leukemias [2]. While modern induction chemotherapy and hematopoietic stem cell transplantation (HSCT) have dramatically improved overall leukemia survival rates, patients diagnosed with MLL-R leukemia have a particularly poor prognosis, with an overall survival less than 50%, and this prognosis is not improved by allogenic HSCT [3]. Thus, novel treatment paradigms must be investigated specifically for MLL-R leukemias, alone and in combination with standard of care (SOC) cytotoxic induction chemotherapeutics.

Great advances have been made in understanding the unique molecular mechanisms that mediate MLL-R driven leukemogenesis, and studies have suggested MLL-R leukemias are largely driven and maintained through epigenetic dysregulation. The histone methyltransferase (HMT) disruptor of telomeric silencing 1-like (DOT1L) has come to the forefront as a critical mediator of MLL-FP mediated leukemogenesis and has been shown to be required for the development, maintenance, and progression of MLL-R leukemias. DOT1L catalyzes the sequential methylation of H3K79 at the promoters of actively transcribed genes and has been strongly implicated as a requisite driver of MLL-FP mediated leukemic transformation and progression. Chromosomal translocations cause the in-frame fusion of MLL most frequently to members of the super elongation complex (SEC)—namely, AF4, AF6, AF9, AF10, and ENL [4]. These oncogenic fusions can interact directly and indirectly with DOT1L, promoting aberrant recruitment to leukemogenic gene promoters like the HoxA cluster or Meis1 [5]. DOT1L causes local H3K79 hypermethylation at these promoters inducing aberrant expression resulting in leukemic transformation [6].

Several independent studies have shown that pharmacologic and genetic inhibition of DOT1L blocks both leukemic transformation and maintenance in vitro and in vivo. Consistent with this paradigm, genetic deletion of DOT1L stops the leukemic transformation of hematopoietic stem cells (HSC) transfected to express MLL fusions, and new small molecule inhibitors of DOT1L are selectively toxic to MLL fusion driven leukemia cells in vitro and in xenograft models [7,8,9,10,11]. However, clinical trials of DOT1L inhibitors have shown underwhelming results with high rates of developed resistance [12]. Given the unique dependence of MLL-FPs on aberrant SAM–dependent activating H3K79 methylation by DOT1L to maintain leukemic potential, we investigated perturbation of Met/SAM metabolism as a novel therapeutic strategy to disrupt abnormal activating epigenetic methylation driving MLL-R leukemias. Targeting Met/SAM metabolism by pharmacological inhibition or dietary restriction may provide a novel adjuvant strategy, which used in combination with DOT1L inhibitors may overcome acquired resistance and improve overall clinical utility of these inhibitors.

The Met/SAM metabolic pathway is essential for cellular trans-methylation (Figure 1A) reactions, including epigenetic gene regulation, as SAM is the primary methyl donor for most cellular methylation reactions, including DNA, RNA, lipids, and histones [13,14]. Additionally, SAM also serves as the sole source of the propylamine moiety required for polyamine biosynthesis [15,16]. SAM is synthesized from methionine by the enzyme methionine adenosyltransferase 2A (MATIIA), and enzyme mediated donation of the SAM methyl group forms the product S-adenosylhomocysteine (SAH). SAH is further metabolized by SAH hydrolase (SAHH) to form homocysteine [17]. Homocysteine, an obligatory intermediate in trans-sulfuration and glutathione synthesis, can then be recycled back into methionine by the enzyme methionine synthase (MTR), which simultaneously converts 5-methyltetrahyrofolate into tetrahydrofolate [18]. Importantly, the intracellular ratio of SAM:SAH dictates the overall methylation potential of the cell, and SAM levels must be maintained in sufficient excess compared to SAH for methylation reactions to proceed. The accumulation of downstream biosynthetic products like SAH, inhibits cellular methylation reactions, including epigenetic methylation of DNA or histones, by impairing cellular methylation potential [19].

We hypothesized that high levels of Met/SAM metabolic flux and expression of the aforementioned enzymatic mediators is required by MLL-R leukemia cells to maintain adequate methylation potential required to enforce aberrant histone methylation and leukemic phenotype. Published literature targeting this pathway specifically in MLL-R leukemia is non-existent, and a very sparse body of work exists for targeting this pathway as a general anti-leukemic therapy, with studies limited to small in vitro studies using established human cell lines and single agent non-specific competitive pharmacological inhibition of MATIIA or SAHH [20]. Here, we show, for the first time, that perturbation Met/SAM metabolism decreases overall methylation potential, deregulates histone methylation dynamics globally and at the DOT1L promoter, decreases DOT1L expression and function, and induces apoptosis in MLL-FP-expressing cells.

## 2. Materials and Methods

### 2.1. Cell Culture and Patient Samples

All established human leukemia cell lines MV411, RS411, and K562 were obtained from American Type Culture Collection (ATCC, Rockville, MD, USA) and cultured in standard RPMI medium supplemented with 10% fetal bovine serum (FBS) and 1% penicillin/streptomycin at 37 °C with 5% CO_2_. Cells were treated with 30µM of 3-deazaadenosine (DZA), a cyclic dinucleotide SAM-binding pocket competitive inhibitor of SAHH, dissolved in DMSO (catalog #9000785, Cayman Chemical, Ann Arbor, MI, USA) in all experiments or cultured in methionine deficient RPMI medium supplemented with 10% FBS and 1% penicillin/streptomycin for experiments involving methionine deprivation. Patient derived xenografts (CCHC-7, CCHC-9, and CCHC-23) were established at Cincinnati Children’s Hospital Medical Center (Cincinnati, OH, USA) from pediatric specimens acquired under an IRB-approved protocol following informed consent at time of relapse. Following engraftment and expansion in NSGS mice, we received the harvested BM aspirates from leukemic mice frozen at –80 °C in RPMI with 10% FBS and 10% DMSO until xenograft. CCHC-7 cells were cultured in vitro in standard RPMI medium supplemented with 10% fetal bovine serum, 1% penicillin/streptomycin, and human cytokines (SCF, FLT3L, TPO, IL-3, and IL-6) at 37 °C with 5% CO_2_.

### 2.2. Annexin V/Propidum Iodide Staining for Apoptotic Cells

Cell death was analyzed and quantified by FACS staining for Annexin V and propidium iodide (PI). Briefly, cells were thoroughly washed twice with ice cold PBS and resuspended in 300ul of 1X Annexin binding buffer. Cells were incubated with 1µL of anti-Annexin V-APC antibody (catalog #640920, BioLegend, San Diego, CA, USA) and 4µL of 1 mg/mL PI solution (Sigma-Aldrich, St. Louis, MO, USA) for 15 min at 4°C, followed by analysis on a Becton Dickinson FACScan using FlowJo software.

### 2.3. Protein Isolation/Quantification and Western Blot Analysis

Protein was isolated from cells in CHAPS lysis buffer and quantified as previously described [21]. Western blot analysis was then conducted as previously described using 30 µg of protein for experiments involving total protein lysates and 15 µg of protein for experiments involving purified histones, using 1:5000 or 1:2000 dilutions, respectively, for primary antibodies, and 1:20,000 dilution of secondary antibodies and proteins of interest were detected by addition of chemiluminescence substrate.

### 2.4. SAM/SAH Reverse Competition ELISA

Intracellular metabolites were isolated on ice by sonication of 10 × 10^6^ cells per timepoint in 1 mL of ice-cold PBS using a 30 kHz sonnicator with probe at 30% amplitude for three 20 s cycles with one minute breaks between. Resultant cell free supernatants were snap frozen and stored at −80°C. Quantification of intracellular SAM and SAH concentration was then conducted using the S-Adenosylmethionine (SAM) and S-Adenosylhomocysteine (SAH) ELISA Combo Kit from Cell Biolabs, INC. (catalog #STA-671-C, San Diego, CA, USA) following the manufacturers’ protocol. The relative methylation potential was then determined for each cell type by expressing the intracellular concentration (ug/mL)/ng protein data as a ratio of SAM to average SAH concentration, normalized to vehicle treated cells cultured in methionine containing media.

### 2.5. Histone Isolation

Histones were isolated from cellular nuclei by acid extraction as follows. Cells were harvested and washed twice in cold PBS and resuspended in Triton Extraction Buffer (TEB: PBS containing 0.5% Triton × 100, 2 mM phenylmethylsulfonyl fluoride (PMSF), 0.02% NaN3) at a cell density of 1 × 10^7^ cells per mL. Cells were lysed by ten minute incubation in TEB buffer at 4 °C with gentle stirring followed by centrifugation at 6500× *g* for ten minutes at 4 °C to pellet the nuclei. The nuclei were resuspended and washed in half the original volume of TEB buffer, followed by centrifugation as before. The nuclei pellet was resuspended in 100 µL of 0.2 N HCl and placed at 4 °C overnight to acid extract histones. The following day debris was pelleted by centrifugation at 6500× *g* for 10 min and the histone supernatant was collected. Following HCl neutralization by addition of 10µL of 2M NaOH, protein content was determined using the BCA assay.

### 2.6. Chromatin Immunopreciptation (ChIP)

Chromatin fixation, isolation, digestion, and immunoprecipitation was performed as previously described in detail [22]. ChIP was performed using anti-H3K4me3 and anti-H3K27me3 antibodies, followed by purification and quantitative RT-PCR as described below, using DOT1L promoter specific primers designed just upstream of putative transcription factor consensus sites predicted by MotifMap software. The DOT1L promoter specific primers were forward primer CCTGGACCGTGACTCTTATGG and reverse primer TCGAATCCCGTCCCAGGAG producing a 99 bp product. RT-PCR was also performed on immunoprecipitated chromatin using control primers against a distal non-specific region approximately 6 kb upstream of the transcription start site (TSS) to ensure results were not technique driven. The control primers were forward primer CACCACGCCCCATCCTATTT and reverse GGGCAGTGAAAAGGGACAGA, producing a 97 bp product (Figure 5A).

### 2.7. RT-PCR

RT-PCR was performed on immunoprecipitated genomic DNA or total isolated RNA reverse transcribed to cDNA as previously described [21]. Primers for qPCR were designed against the promoter sequence or transcribed sequence using Primer Express 3.0 software (Applied Biosystems, Foster City, CA, USA) per the manufacturer’s instructions for SYBR green dye assays. The DOT1L mRNA primers were forward primer CCAACACGAGTGTTATATTTGTGAA and reverse primer TGAGTTTTGGGTTTTTCAGACTAGA, producing a 306 bp product. RT-PCR was performed using iTaq Universal Sybr Green Supermix (Bio-Rad), and relative expression levels or relative histone methylation occupancy was analyzed using ΔΔCT method. Data was normalized to 12.5% total chromatin inputs for RT-PCR following ChIP or β-actin expression for experiments on reverse transcribed mRNA. PCR reactions were analyzed on a BioRad CFX96 using BioRad CFX Manager 3.1 software (Hercules, CA, USA).

### 2.8. Clinical Chemotherapeutics

Clinical formulations of both doxorubicin and cytarabine were obtained from the James Graham Brown Cancer Center Pharmacy as self-sealing vials containing 20 mg/10 mL or 2g/20mL respectively dissolved in saline, manufactured by APP a division of Fresenius Kabi USA LLC (Lake Zurich, IL, USA).

### 2.9. In Vivo Xenograft Studies

As previously described NRGS (NOD/RAG1/2^−/−^IL2Rγ^−/−^Tg[CMV-IL3,CSF2,KITLG]1Eav/J, stock no: 024099) mice producing 2-4 ng/mL of human IL-3, GM-CSF, and SCF [23] were obtained from Jackson laboratories (Bar Harbor, ME, USA) and bred and maintained under standard conditions in the University of Louisville Rodent Research Facility (Louisville, KY 40202, USA) on a 12-h light/12-h dark cycle with food and water provided ad libitum. For xenograft studies, NRGS mice received 1.25 × 10^5^ human cells (CCHC-7 or CCHC-9) suspended in 200µl PBS by bolus IV injection and *n* = 5 for all study cohorts. Mice then received an isovolumetric IV bolus of the vehicle (PBS) or were treated with our previously described high intensity 5 + 3 induction regimen with or without 25 mg/kg DZA dissolved in 200µl of PBS [24]. Animal studies are done in accordance with IACUC #18256 approved 10/21/2018.

### 2.10. Statistical Analysis

All statistics were performed using GraphPad Prism 8 software. Unless specified below, significance was determined by one-way ANOVA, followed by Tukey tests, using a cut off of *p* < 0.05. For all survival curves, the log rank (Mantel-Cox) test was used, with a cut off of *p* < 0.05.

## 3. Results

### 3.1. Alteration of Met/SAM Metabolism Impairs Cellular Viability and Induces Apoptosis in MLL-R Cell Lines

Cellular Met/SAM metabolism was disrupted, either by deprivation of exogenous methionine in the culture media, or pharmacologic inhibition of downstream metabolism using the competitive SAH hydrolase (SAHH) inhibitor 3-deazaadenosine (DZA), in two MLL-AF4 expressing cell lines (MV411, RS411), MLL-AF6 expressing patient blasts (CCHC-7) and BCR-ABL driven K562 cells. DZA mediated inhibition of Met/SAM metabolism induced a dose dependent impairment of cellular viability as quantified by Alamar Blue assay selectively in MLL-R cells, but not MLL-R independent K562 cells (Figure 1B, left). Reducing the concentration of exogenous methionine also impairs cellular viability, but more potently in the MLL-R cells compared to K562 cells (Figure 1B, right). Using the data obtained from both dose curves we determined an appropriate concentration of DZA and timeframe to investigate methionine deprivation. Annexin V staining revealed that methionine deprivation and/or inhibition of SAH metabolism potently induced apoptosis MV411 cells with the combination producing an additive increase in apoptosis (Figure 1C, left). Interestingly, RS411 cells only underwent apoptosis in the presence of DZA, but additive effects were still observed with simultaneous methionine deprivation and DZA treatment (Figure 1C, middle). K562 cells lacking MLL-FP expression, were resistant to apoptosis induction through either methionine deprivation or inhibition of downstream SAHH mediated metabolism (Figure 1C, right). Consistent with apoptosis induction, increased PARP-1 and caspase-3 cleavage was detected in both MLL-AF4 expressing cell lines following perturbation of Met/SAM metabolism, and in agreement with Annexin V staining, cleaved caspase-3 was detected in RS411 cells only under conditions of DZA-mediated SAHH inhibition (Figure 1D).

### 3.2. Perturbation of Met/SAM Metabolism Increases Intracellular SAH, Decreases Overall Methylation Potential, and Alters Global Histone Methylation Dynamics in MLL-R Cell Lines

Counterintuitively, disruption of Met/SAM metabolism did not significantly alter intracellular SAM concentrations in any of the cell lines tested (Figure 2A), suggesting that these cells exhibit some capacity to recycle Met/SAM from other intracellular metabolites. MV411 cells showed a significant increase in intracellular SAH concentration 24 h post exposure to all experimental conditions (Figure 2B, left). RS411 cells also showed a significant increase in intracellular SAH but only after exposure to DZA in methionine containing conditions (Figure 2B, right). Finally, we determined the overall cellular methylation potential by expressing the data as a ratio of SAM:SAH concentration normalized to vehicle treated cells in methionine containing media. Methionine deprivation or DZA exposure significantly lowered methylation potential in MV411 cells (Figure 2C left), and only in RS411 cells treated with DZA with or without exogenous methionine (Figure 2C, right). Western blot analysis of purified histone extracts from both MLL-R cell lines showed a loss of the common activating modification H3K4me3, and an increase in the repressive H3K27me3 mark upon DZA mediated SAHH inhibition. Most promisingly, perturbation of Met/SAM metabolism in either MLL-R cell line potently reduced global levels of H3K79me2, the DOT1L catalyzed activating modification absolutely required by MLL-R leukemia for survival and maintenance of malignant potential (Figure 2D).

### 3.3. Disruption of Met/SAM Metabolism Reduces mRNA Expression and Protein Levels of the H3K79 Methyltransferase DOT1L and Induces DNA Damage

DOT1L mRNA expression was significantly reduced in MV411 cells following 24 or 36 h exposure to experimental conditions (Fig 3A, left). Consistent with these findings DOT1L protein levels were also diminished and interestingly DNA damage was observed under conditions of DOT1L depletion as determined by increased pH2.AX (Figure 3A, right). In alignment with our cytotoxicity data, a decrease in DOT1L mRNA was only observed upon DZA treatment of RS411 cells 24 h post experiment initiation, but by 36 h, DOT1L mRNA levels are decreased under all conditions (Figure 3B, left). Protein expression of DOT1L was reduced only upon exposure to DZA 24 h post treatment, and under all experimental conditions by 48 h, DNA damage was observed by pH2.AX foci formation (Figure 3B, right). Conversely, DOT1L mRNA expression was significantly increased or remained unchanged in non-MLL rearranged K562 cells, but DOT1L protein levels were still reduced (Figure 3C). Finally, to further strengthen correlation between Met/SAM metabolic status, DOT1L expression, and apoptosis induction, we either DZA treated RS411 cells for 24 h followed by washout or deprived MV411 cells of methionine for 48 h followed by re-plating in methionine rich media. Exposure to DZA for 24 h or methionine deprivation for 48 h decreased DOT1L expression and induced caspase-3 cleavage, followed by a subsequent increase in DOT1L expression and decrease in caspase-3 cleavage during recovery in drug free or methionine rich media (Figure 3D,E).

### 3.4. Decreased DOT1L Expression is Correlated to Changes in Histone Methylation Dynamics at the DOT1L Promoter

Chromatin immunoprecipitation (ChIP) was conducted on MV411 and RS411 cells under all experimental conditions using an antibody against H3K27me3, followed by RT-PCR using DOT1L promoter specific primers. Promoter-specific primers were designed against a DOT1L promoter region just upstream of two putative transcription factor binding sites, as predicted by MotifMap software. Control primers against an arbitrary distal location were designed as an internal control to ensure that results were not due to technical bias (Figure 4A). We found that inhibition of SAH metabolism and/or deprivation of exogenous methionine significantly elevated levels of the repressive H3K27me3 modification at the DOT1L promoter in MV411 cells 24 h and 48 h post exposure to experimental conditions, with an even greater increase in DOT1L promoters H3K27me3 48 h post combined methionine deprivation and DZA treatment (Figure 4B). As we expected, RS411 cells only showed increased levels of DOT1L promoter H3K27me3 upon DZA mediated inhibition of SAH metabolism, again matching with our previous cytotoxicity and expressional data in RS411 cells (Figure 4C). Our results suggest that sufficient methylation potential and Met/SAM metabolic flux are necessary to maintain required levels of DOT1L expression in MLL-R cells.

### 3.5. Patient-Derived MLL-R Leukemic Blasts are Sensitive to Alterations in Met/SAM Metabolism and Show Corresponding Changes in Global and Promoter-Specific Histone Methylation

We next examined if similar changes in histone methylation dynamics and DOT1L expression and function were observed in patient-derived MLL-AF9-expressing blasts (CCHC-7). CCHC-7 cells grow under standard culture conditions with supplementation of human hematopoietic cytokines, we therefore plated these cells under the previous experimental conditions and similarly examined apoptotic effects, alterations to global histone methylation dynamics, and changes in DOT1L expression. Like MV411 and RS411 cells, CCHC-7 cells undergo apoptosis, as determined by immunoblotting for caspase-3 and PARP-1 cleavage (Figure 5A). However, the cytotoxic effects of perturbation of Met/SAM metabolism in CCHC-7 cells more closely resemble RS411 cells, in that only DZA induced inhibition of SAHH with or without methionine resulted in rapid apoptosis induction. Furthermore, we observed a decrease in overall DOT1L protein expression and similar reduction in global H3K79me2 methylation in the patient derived CCHC-7 cells, as compared to established MLL-R cell lines (Figure 5B). Interrogation of the DOT1L promoter by ChIP using a H3K27me3 specific antibody followed by RT-PCR using DOT1L promoter specific primers, showed a significant increase in H3K27me3 occupancy at the DOT1L promoter in CCHC-7 cells (Figure 5C). These results are highly similar to our in vitro studies utilizing the established MLL-R cell lines, we therefore felt confident in moving forward to in vivo studies.

### 3.6. Pharmacologic Inhibition of SAH Metabolism Significantly Prolongs the Survival of MLL-R Xenograft–Bearing Mice, in Combination with SOC Induction Therapy

We finally examined the in vivo efficacy of DZA mediated SAHH inhibition in a highly advanced and clinically relevant xenograft model of patient MLL-R leukemia. Cohorts of transgenic NRGS mice (*n* = 5 per group) were xenografted with three pediatric patient AML samples, each bearing a different MLL-R. Seven days post xenograft, the mice were given isovolumetric bolus intravenous injections of either saline, induction chemotherapeutics, or induction chemotherapeutics combined with 25 mg/kg DZA in a clinically similar and previously defined 5 + 3 regimen [24]. Combined treatment using 5 + 3 induction and DZA significantly prolonged the survival of MLL-R xenograft–bearing NRGS mice, compared to induction alone (Figure 5D). Combinatorial therapy of mice harboring CCHC-9 leukemia significantly prolonged their survival, and two of five mice in the combination cohort survived to the study endpoint without developing disease (Figure 5D, right).

## 4. Discussion

Modern, aggressive cytotoxic chemotherapy has allowed for high rates of durable remission across a variety of leukemias. However, given the inherently heterogenous nature of leukemic disease, certain leukemia sub-types, like those bearing MLL-R, remain poorly treated and have the lowest overall survival rates. Furthermore, MLL-R is detected in 10% of all cases of leukemia and 80% of all pediatric leukemias, highlighting the need for novel, highly efficacious therapeutic paradigms specifically for MLL-R leukemia. To address this need we investigated a new therapeutic strategy targeting the now well understood underlying epigenetic mechanisms driving MLL -R mediated leukemogenesis. In short, MLL-FPs gain oncogenic activity through mis-localization of the transcriptionally activating H3K79 methyltransferase DOT1L and the fusion partner associated SEC complex, to developmental gene promoters (HOXA cluster, MEIS1, etc.) driving their aberrant expression and subsequent leukemic transformation and progression. MLL-R has been defined as a unique “epigenetic lesion” dependent on high levels of aberrant DOT1L-mediated H3K79 methylation to maintain enforced leukemic gene expression and phenotype. Therefore, we examined if perturbation of Met/SAM metabolism, which controls both the availability of the methyl donor SAM and overall cellular methylation potential, could be a novel therapeutic paradigm specifically for leukemias bearing MLL-R.

As we hypothesized, perturbation of Met/SAM metabolism by deprivation of exogenous methionine or inhibition of downstream SAH metabolism was cytotoxic and induced apoptosis in established MLL-AF4 bearing AML (MV411) and ALL (RS411) cell, but not BCR-ABL-driven K562 cells or normal murine BM cells or normal human PBMCs (data not shown). However, methionine deprivation reduced relative cellular viability in all cells tested by Alamar Blue assay, but to a larger degree in the MLL-R cells. The observed loss of relative viability, even in K562 cells, reflects a reduction in cellular proliferative capacity induced by nutrient deprivation and not necessarily cell death. Furthermore, disruption of Met/SAM metabolism also resulted in the induction of DNA damage in both MLL-R cell lines, but also in K562 cells under conditions of methionine deprivation, perhaps through decreased polyamine biosynthesis required for efficient DNA damage repair [25,26,27]. DOT1L is also directly involved in DNA damage repair and thus loss of its expression may also contribute to the accumulation of DNA damage. Differential effects in cytotoxicity were observed between the MLL-R cell lines, with RS411 cells only undergoing apoptosis upon inhibition of SAH metabolism but not methionine deprivation alone, yet DZA-mediated apoptotic effects were still potentiated by methionine deprivation. These findings may indicate inherent cell type specific differences in cellular metabolism likely dictate sensitivity to methionine deprivation. For example, like many tumors RS411 cells lack expression of methylthioadenosine phosphorylase (MTAP), an enzyme required for interconversion of the polyamine biosynthetic product 5-MTA back to methionine, resulting in accumulation of 5-MTA another feedback inhibitor of trans-methylation and polyamine biosynthetic reactions [28]. Therefore, RS411 cells likely maintain relatively large intracellular pools of SAM and decarboxylated SAM, thereby allowing trans-methylation and polyamine biosynthesis to progress [29] and, due to this cell-intrinsic difference, show limited sensitivity to methionine deprivation in the short timeframe examined. Longer exposure to conditions of methionine deprivation would likely induce apoptosis even in RS411 cells as evidenced by a reduction in relative cellular viability and increased PARP-1 proteolysis, which can occur independently of caspase-3 activation through cleavage by caspases 1, 7, cathepsins, or TGF-β signaling [30,31,32].

Furthermore, perturbation of Met/SAM metabolism also caused significant changes in the intracellular metabolite pool of SAH, but interestingly not SAM, indicating that these cells maintain some capacity to recycle or intraconvert methionine and SAM from other metabolite sources including homocysteine and 5-MTA [33]. Additionally, recent work from several groups has shown that methionine, and by extension SAM, can also be readily salvaged through cellular autophagy and may explain the lack of significant change in intracellular SAM concentration [34,35,36]. Our data indicate that perturbation of Met/SAM metabolism significantly reduces overall cellular methylation potential in the tested MLL-AF4 expressing cell lines globally altering histone methylation dynamics involved in epigenetic regulation of gene expression. Furthermore, in line with our cytotoxicity data, either cell line undergoes apoptosis only under conditions where cellular methylation potential is significantly decreased, and our data indicates more pronounced rapid cytotoxic effects upon DZA-mediated SAHH inhibition. Our data suggests that perturbation of Met/SAM metabolism deregulates intracellular methylation potential and is toxic to MLL-R-bearing cells, one likely mechanism being through disruption of DOT1L expression through increased promoter H3K27me3 and subsequent loss of H3K79 methylation–mediated leukemogenic gene expression (Figure 6). While it would be expected that disruption of SAM/Met metabolism would reduce all histone methylation, we show a global and DOT1L promoter specific increase in H3K27me3. Our findings suggest that under conditions of reduced methylation potential in MLL-R cells, the enzymatic effects of certain HMTs predominate, likely based on their intrinsic affinity for SAM or conversely their sensitivity to feedback inhibition by downstream metabolites, like SAH [37,38,39].

Finally, we confirmed our findings using patient-derived AML cells bearing MLL-R in vitro and tested in vivo efficacy along with SOC induction therapy in an advanced xenograft model. Patient-derived CCHC-7 cells undergo apoptosis and show cell line comparable changes in histone methylation dynamics and DOT1L expression, and our findings indicate pharmacologic inhibition of SAHH may be the most attractive therapeutic target. The addition of 25 mg/kg DZA to a clinically similar 5+3 regimen of SOC cytotoxic induction, significantly enhanced the survival of mice xenografted with three separate relapsed pediatric AMLs, each bearing a unique MLL-R, when compared to induction alone. These findings are especially promising, considering DZA is a relatively poor competitive SAHH inhibitor (K_i_ = 3.9 µM) with a short half-life (23 min). Our data suggests that targeted inhibition of Met/SAM metabolism may provide a unique therapeutic paradigm capable of deregulating the underlying epigenetic mechanisms that drive all MLL-R leukemias, thereby improving sensitivity and therapeutic response to cytotoxic chemotherapy.

Unfortunately, no clinically useable highly specific non-competitive allosteric inhibitors currently exist for the enzymatic mediators of Met/SAM metabolism (MATIIA, SAHH, MTHR, etc.), and small molecule DOT1L inhibitors have demonstrated lackluster performance in clinical trials with high rates of accompanying drug resistance. Data from our studies support the continued investigation of novel small molecule trans-methylation inhibitors—or manipulation of dietary methionine—specifically for the therapy of MLL-R leukemia. Such strategies may be able to improve the efficacy of existing DOT1L inhibitors and/or reduce rates of acquired resistance, thereby improving clinical utility of such small molecule inhibitors. Furthermore, disruption of SAM/Met metabolism, especially by dietary restriction, may be an immediate alternative strategy to improve SOC chemotherapy outcomes in patients with MLL-R leukemia, as DOT1L inhibitors have yet to be FDA approved and mass produced. In conclusion, our findings reveal a previously unexplored metabolic vulnerability in aggressive MLL-R leukemia, which could be exploited in the future to improve therapeutic outcomes and overall patient prognoses.

## Figures and Tables

**Figure 1 cells-08-01322-f001:**
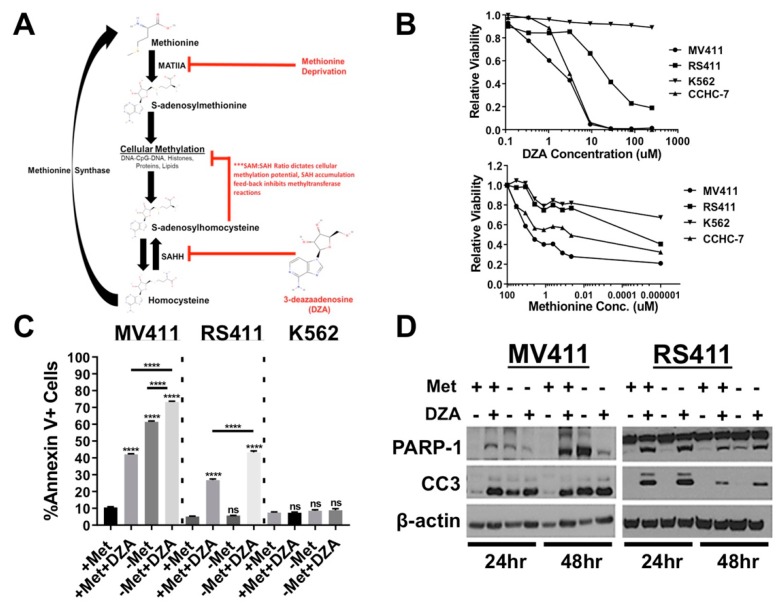
Perturbation of methionine (Met)/S-adenosylmethionine (SAM) metabolism potently induces apoptosis and reduces cell viability in established mixed lineage leukemia rearrangement (MLL-R) leukemia cell lines. (**A**) Simplified schematic of Met/SAM metabolism showing important enzymatic mediators, substrates and products. Deregulation of Met/SAM metabolism and subsequently methylation potential was achieved by targeting two distinct nodes, either by reducing synthesis of the methyl donor moiety SAM by methionine deprivation or pharmacological inhibition of downstream SAH metabolism and methionine recycling. (**B**) Dose and time dependent reduction in relative survival of MLL-AF4 expressing MV411 (AML) and RS411 (ALL) cells treated with increasing concentrations 3-deazaadenosine (DZA) as quantified by Alamar Blue vitality assay 48 h post treatment. K562 cells lacking MLL-fusion expression show no reduction in relative survival upon DZA treatment (left). MLL-R cells also show a greater reduction in cell numbers upon methionine withdrawal as compared to K562 cells (right). (**C**) From left to right; Methionine deprivation or exposure to 15µM DZA potently induced apoptosis at 48 h in MV411 cells singly, and the combination showed an additive increase in apoptosis as determined by Annexin V+ staining. RS411 cells only undergo apoptosis upon exposure to 15µM DZA, and this effect is amplified by methionine deprivation. K562 cells lacking MLL-R show no induction of apoptosis 48 h post DZA exposure or methionine deprivation (Tukey HSD, *p* < 0.0001). (**D**) Changes in protein expression corresponding with apoptosis induction (PARP-1 and Caspase-3 cleavage) were observed under all experimental conditions in MV411 cells (left), while RS411 cells only undergo apoptosis upon DZA-mediated SAHH inhibition (right).

**Figure 2 cells-08-01322-f002:**
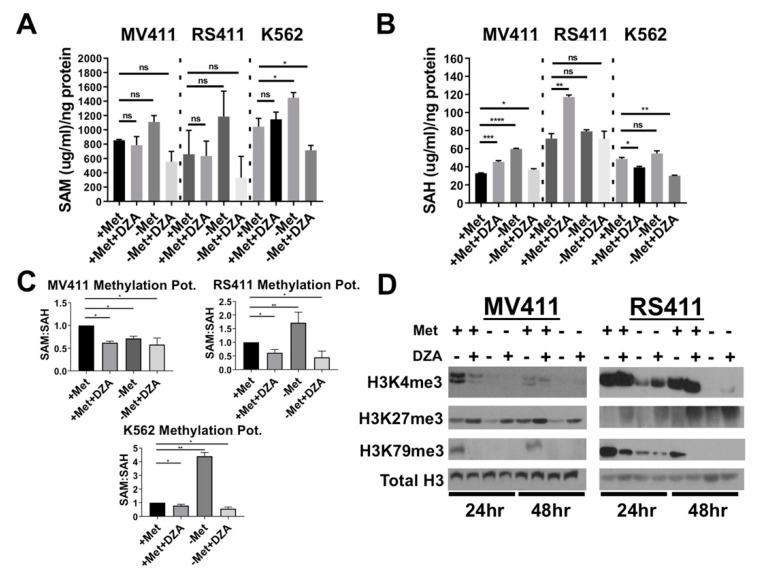
Disruption of Met/SAM metabolism increases intracellular SAH concentration and decreases overall methylation potential. (**A**) From left to right; Intracellular SAM concentration was not significantly changed in either MV411 or RS411 cells but was significantly elevated in K562 cells upon methionine starvation and significantly reduced with DZA treatment and methionine deprivation (Tukey HSD, * = *p* < 0.05). (**B**) From left to right; Intracellular SAH concentration was increased in MV411 cells under all experimental conditions and timepoints, while only DZA-mediated inhibition of SAH metabolism sufficiently increased intracellular SAH concentration in RS411 cells (Tukey HSD, *= *p* < 0.05 ** = *p* < 0.01, *** = *p* < 0.001, **** = *p* < 0.0001). (**C**) Overall cellular methylation potential (SAM:SAH) was significantly reduced in MV411 cells 24 h post exposure to experimental conditions. RS411 cells only show significant reduction in methylation potential following 24 h exposure to DZA, and conversely, a significant increase in methylation potential was observed following 24 h of methionine deprivation (Tukey HSD, * = *p* <0.05 ** = *p* <0.01, *** = *p* <0.001, **** = *p* <0.0001). (**D**) Global histone methylation dynamics are altered in both MLL-AF4 expressing cell lines upon disruption of Met/SAM metabolism resulting in a reduction in the activating H3K4me3 modification and an increase in the repressive H3K27me3 following DZA-mediated SAHH inhibition. Importantly, perturbation of Met/SAM metabolism potently reduced global levels of DOT1L-dependent H3K79me2, an activating modification absolutely required by MLL-R leukemic cells.

**Figure 3 cells-08-01322-f003:**
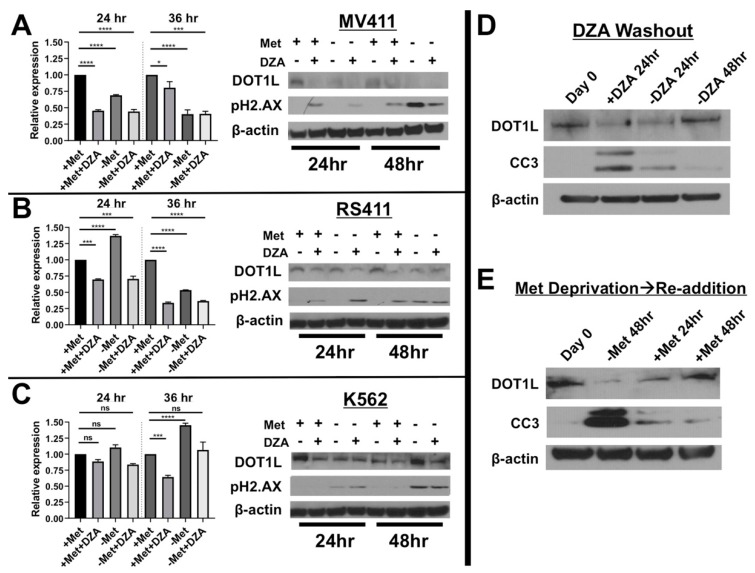
**Alteration of Met/SAM metabolism reduces mRNA and protein expression of the required H3K79 methyltransferase DOT1L selectively in MLL-R leukemia cells.** (**A**) DZA treatment or methionine deprivation of MV411 cells significantly reduced DOT1L mRNA (Tukey HSD, * = *p* < 0.05 *** = *p* <0.001 **** = *p* <0.0001) following 24 or 36 h exposure to experimental conditions (left) and these changes are reflected in DOT1L protein levels (right). Reduction in DOT1L expression also correlated with the induction of DNA damage as determined by pH2.AX formation, likely due to the critical role DOT1L plays in regulation of the DNA damage repair response. (**B**) RS411 cells display a significant reduction in DOT1L mRNA (Tukey HSD, *** = *p* <0.001 **** = *p*< 0.0001) and protein levels (right) following DZA mediated SAHH inhibition, but not with methionine deprivation alone at 24 h. By 36 h methionine deprivation alone significantly reduced DOT1L mRNA (Tukey HSD, **** = *p*< 0.0001) and protein expression in RS411 cells, and again DNA damage induction was detected only under conditions of DOT1L reduction. (**C**) Non-MLL-R K562 cells lacking functional dependence on DOT1L show no significant changes in DOT1L mRNA expression 24 h post study initiation, however DOT1L mRNA was significantly reduced 36 h after DZA exposure, but was significantly increased following 36 h of methionine deprivation (Tukey HSD, *** = *p* < 0.001 **** = *p* < 0.0001). Interestingly, modulation of DOT1L protein levels was observed even in K562 cells (right), and pH2.AX foci formation was also detected only under conditions of methionine deprivation in K562 cells. These results may suggest that appropriate regulation of Met/SAM metabolism and one-carbon sensing may directly play a more universal role in regulation of DOT1L expression and function in multiple cell types, regardless of MLL-R mediated functional dependence on DOT1L and H3K79 methylation. (**D**) To further correlate SAM/Met metabolism with DOT1L expression RS411 cells were cultured for 24 h in the presence of DZA after which cells were washed and replated in DZA free, methionine containing media. Total protein lysates were collected at the indicated timepoints, and immunoblotting revealed that DOT1L expression was decreased following DZA treatment with an increase in caspase-3 cleavage. Washout of DZA increased DOT1L expression back to initial levels, with a corresponding reduction in cleaved caspase-3. (**E**) MV411 cells deprived of methionine for 48 h had reduced DOT1L protein expression and increased caspase-3 cleavage. Reintroduction of methionine in the media rescued DOT1L expression to initial levels, with a corresponding reduction in caspase-3 cleavage.

**Figure 4 cells-08-01322-f004:**
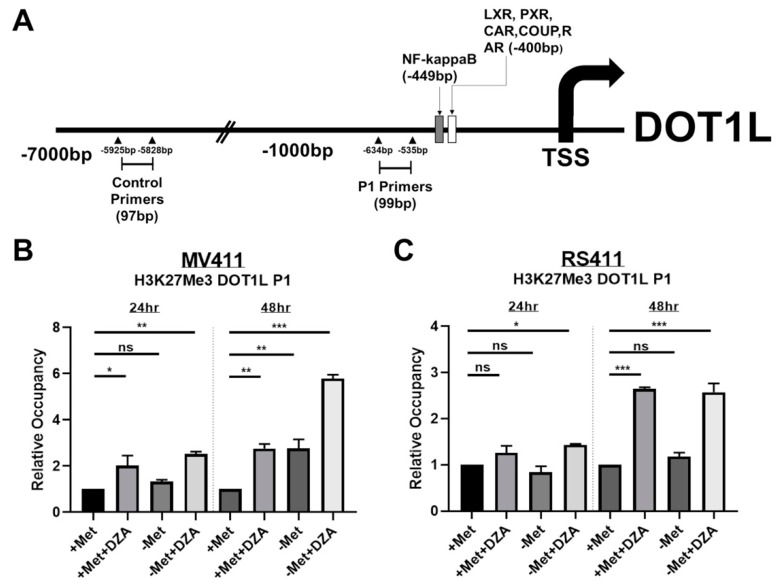
Decreased expression of DOT1L is correlated to changes in histone methylation dynamics specifically at the DOT1L promoter. (**A**) DOT1L promoter map illustrating putative transcription factor binding sites as determined by MotifMap software, as well as the specific binding sites of the DOT1L promoter specific primers (P1) and the distal control primers. (**B**) MV411 cells show significantly elevated levels of H3K27me3 occupancy at the DOT1L promoter following DZA treatment or methionine deprivation and by 48 h an additive increase was observed under simultaneous methionine deprivation and DZA-mediated SAHH inhibition (Tukey HSD, *= *p* <0.05 **= *p* <0.01 ***= *p* <0.001). (**C**) RS411 cells exhibit significantly increased DOT1L promoter H3K27me3 occupancy following 24 h of simultaneous methionine deprivation and DZA exposure, and by 48 h, DZA treatment alone but not methionine deprivation alone is sufficient to significantly elevate DOT1L-promoter H3K27me3 occupancy (Tukey HSD, *= *p* <0.05 ***= *p* <0.001).

**Figure 5 cells-08-01322-f005:**
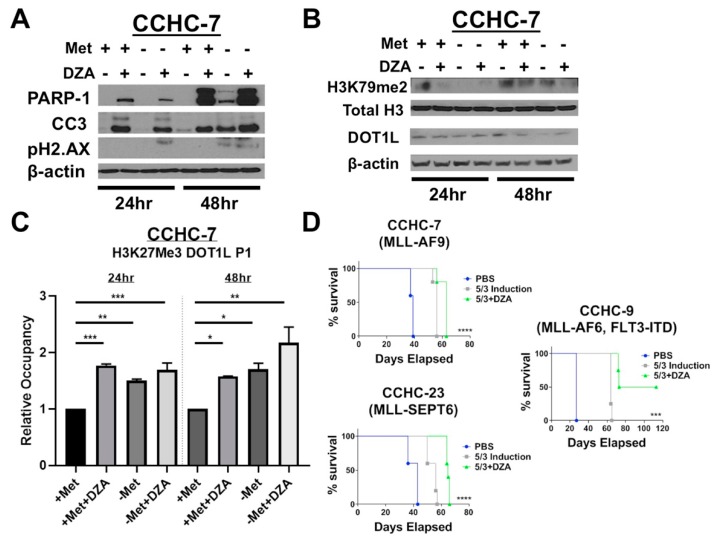
Patient-derived MLL-R leukemic blasts are sensitive to perturbation of Met/SAM metabolism and show changes in global and DOT1L promoter–specific histone methylation dynamics consistent with tested MLL-R cell lines. Combining DZA with 5 + 3 induction therapy significantly prolonged the life of mice harboring several different MLL-R patient derived xenografts. (**A**) Patient-derived CCHC-7 cells behave similarly to the RS411 cell line and only undergo apoptosis following DZA-mediated SAHH inhibition with or without methionine deprivation, as determined by PARP-1 and caspase-3 cleavage. Methionine deprivation for 48 h was also sufficient to induce apoptosis and PARP-1 and caspase-3 cleavage. (**B**) CCHC-7 cells show a profound early loss in global H3K79me2 levels following 8 h of exposure to experimental conditions, and this reduction was more modest 24 h post exposure to either DZA or methionine deprivation alone. However, the combination of simultaneous DZA mediated SAHH inhibition and methionine deprivation was still sufficient to decrease global H3K79me2 even after 24 h (top). DOT1L protein expression was also reduced in CCHC-7 cells by methionine deprivation and/or DZA treatment (bottom). (**C**) CCHC-7 cells display significantly elevated levels of DOT1L promoter H3K27me3 occupancy 24 or 48 h after exposure to all experimental conditions (Tukey HSD, *= *p* <0.05 **= *p* <0.01 ***= *p* <0.001). (**D**) NRGS mice (*n* = 5 for all cohorts in all studies) were xenografted with 1.25 × 10^5^ CCHC-7, CCHC-23, or CCHC-9 patient leukemic cells each bearing a unique MLL-R, and seven days post xenotransplant, mice began bolus intravenous infusion of isovolumetric amounts of 5 + 3 induction therapy (3 mg/kg doxorubicin, days 1–3 and 75 mg/kg cytarabine, Days 1–5), 5 + 3 induction plus 25 mg/kg DZA (days 1–5), or the vehicle PBS alone. DZA in combination with 5 + 3 induction significantly prolonged the life of these mice as compared to either 5 + 3 induction therapy or vehicle treatment alone in all xenograft studies regardless of particular patient MLL-R (Mantel-Cox, *** = *p* <0.001 *** *= *p* <0.0001), and promisingly, two of the five mice bearing CCHC-9 xenografts completely failed to develop leukemic disease and survived till the endpoint of our study (far right).

**Figure 6 cells-08-01322-f006:**
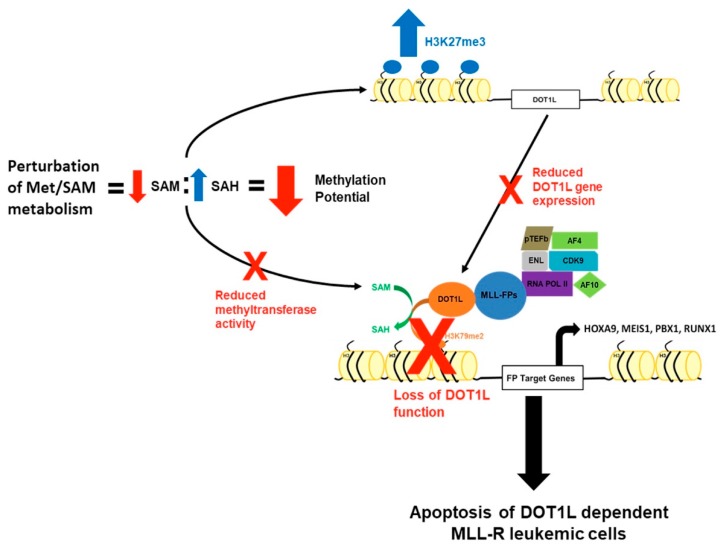
A mechanism by which perturbation of Met/SAM metabolism induces cytotoxicity and apoptosis in MLL-R-bearing cells. Perturbation of Met/SAM metabolism, either by deprivation of exogenous methionine or by pharmacologic inhibition of downstream SAH metabolism, results in a significant increase in the levels of intracellular SAH leading to an overall decrease in global cellular methylation potential. Under conditions of decreased methylation potential, there is a global reduction in methyltransferase activity and a DOT1L promoter–specific increase in the repressive H3K27me3 modification, resulting in decreased DOT1L expression and function. MLL-R leukemic cells are uniquely dependent on DOT1L expression and function to maintain aberrant leukemogenic gene expression and survival, and loss of DOT1L function and expression may partially explain their increased sensitivity to alterations of Met/SAM metabolism.

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
