# Peer review of "Perturbation of Methionine/S-adenosylmethionine Metabolism as a Novel Vulnerability in MLL Rearranged Leukemia"

_cells, 2019, doi:10.3390/cells8111322_

Round 1

Reviewer 1 Report

The manuscript is of good quality and I support it's publication.

Author Response

Thank you so much for finding our work important and acceptable for publication.

Reviewer 2 Report

The work from Barve et al. aims at disrupting the methionine/S-adenosylmethionine metabolic pathway in a particular subset of leukemia, driven by MLL translocation. Literature data show that this subgroup is particularly addicted to DOT1L methylation activity, but the attempts to target this enzyme are thus far inconclusive. So authors suggest a novel approach to target this addiction, by means of metabolic targeting the substrates of the methylation process.

Overall, the paper is well written, the founding hypothesis appears scientifically sound, and the experiments carried out to support this idea seems fairly convincing.

However, there are a few issues that need to be addressed, in order to make this paper suitable for publication in this journal.

It is important to extend the study to the evaluation of normal cells which is absent in this report. In fact, results on normal progenitor should must be be included, showing how normal cell would react to the disrupting of a such important metabolic pathway, especially in light of the epigenetic impact that such inhibition could exert. Authors should comment on the inhibitor (DZA) choice and characteristics, as it is relegated to the material and methods section. Does the methionine deprivation imply that there is no methionine in the growth medium? It must be clearly stated. If that’s the case, it seems to me that there’s a discrepancy between number of RS411 cells in figure 2b (cell count very low), and annexin V level of the –Met condition (AnnV+ cells very few). Please add a comment to explain. Not only SAM doesn’t decrease in methionine deprivation condition, but it appears slightly increased in all cell line (Fig 2A). This is in contrast with what expected, and fairly important in the paper credibility. Did the author try a different technique to measure SAM and SAH levels? Or did they measure authophagy to give support to what stated in the paper (methionine scavenging by authophagy)? Furthermore, it is unclear how authors have calculated the SAM/SAH ratio (Fig 2C), please provide a better explanation and the numerical values, cause it seems not in agreement with the previous two graphs, showing SAM and SAH levels. Figure 3D: the description text under the figure is missing. There are no references in the discussion section

Reviewer 3 Report

This article from Barve et al. describe the effect of pharmacological inhibition of Methionie/SAM metabolism in the proliferation and survival of cells carrying an MLL fusion proteins and their effectiveness as a new therapeutic strategy for the treatment of MLL associated leukemias. The authors try to establish the links between the negative effects on survival observed in MLL cell lines upon treatment with DZA or methionine deprivation, overall changes in histone methylation and DOT1 expression.

Whereas these results are potentially interesting for other scientists in the field, there are currently some major issues that preclude its publication in the actual form.

Major comments

1.- It is surprising that after 48 hr of methionine deprivation in the RS411 cell line no apoptosis can be measured by Annexin V, specially taking into account that levels of PARP are clearly upregulated in these conditions. Also PAPR expression levels in the MV411 cell line do not reflect the additive effect of both treatments. The authors should comment on this.

2.-The authors claim that “perturbation of Met/SAM metabolism …..decreases overall methylation potential and alters global histone methylation dynamics.”. Since SAM is the main donor of methyl groups for both DNA and Histone methylation, it is difficult to understand, how a reduction in the SAM levels, either through methionine deprivation of by pharmacologically block the synthesis pathway can result in both the increase of methylation marks at the DOT1 promoter (H3K27m3) and the decrease in global H3K79 methylation described by the authors in Figure 2. Wouldn´t both processes be equally affected by the reduction in SAM levels? This should be clearly explained.

3.- Also in relation with the previous point, results showing the effects of methionine deprivation or DZA on global histone methylation are not clear. Whereas at 24 and 48 hours a clear reduction in the levels of H3K4me3 is detected in the MV411 cell line, results in the RS411 are inconclusive. Besides, the results obtained with H3K27me3 are not clear in methionine starvation conditions. Authors should at least provide a quantification of these results, for example, measuring band intensity. A more desirable quantification method would involve chromatin immunoprecipitation of these histone marks at 3/4 different loci in the conditions tested.

4.- If the effect seen in the cells lines analyzed are directly mediated by a reduction in the H3K79m3 mark linked to DOT1 activity, these results would have to be comparable to those exerted by DOT 1 inhibitors. Why weren´t these inhibitors used as control?

5.- Regarding the DNA damage experiments, the authors seem to overlook the fact that H2.AX was also significantly upregulated in the K562 cell line, and levels of this marker seem to increase even further in this cell line than in the RS411 cell line at the 48 hours timepoint. It seems clear that both methionine starvation and DZA treatment increase DNA damage in these cells, indicating that this effect is somewhat pleiotropic, which is obviously worrisome taking into account that these treatments are being tested as a possible therapeutic strategy.

These results also seem to clash with those showing a minor effect of the DZA treatment on the apoptosis of the K562 line compared to the RS411 showed in figure 1. Could the authors comment on this?

6.- Chromatin IP experiments are not adequately performed. At least three different parts of the promoter should be analyzed for occupancy of H3K27m3 in the same experiment to be able to argue solid changes in this mark. Besides, it is crucial to know how the results are obtained. The information in the materials and methods takes you to a chain of articles but there is no information there regarding the normalization. How was the normalization performed? This should be explained in detail since in histone marks analysis is crucial to take into account the underlying distribution of histones.

7.- Some small molecules that specifically inhibit DOT1 have already been published. What would be the advantage of using the approach described by the authors over the use of DOT1 inhibitors, since the effect of these seem somehow pleitropic?

Minor comments

1.- The overall quality of the images is an important issue. This should be corrected since even the labels of the samples/conditions tested in the different experiments are hard to see. Image resolution must be increased.

2.- The same nomenclature should be maintained throughout the paper. As an example, in Figure 1, the conditions where only DZA has been added are labeled as +DZA in figure 1C and as +Met+DZA in figure 1D. Please, be consistent.

3.- Although CCHC-7 cell line was initially used to test the effect of Met reduction and DZA, this line was not used for further experiments in spite of the fact that the CCHC-7 cell line showed in these experiments a similar response to that of the MV411 and RS411 cell lines. Could the authors explain why? And also, if this line is not going to be used further it should be deleted for the results in figure 1B.

Round 2

Reviewer 3 Report

This reviewer has no comments for the authors.

Author Response

thank you.  we did upload all new versions of figures with the last revision, but for some reason they did not make it into the version of the manuscript.  we are attempting to rectify this situation.